# PTEN Loss Enhances Error-Prone DSB Processing and Tumor Cell Radiosensitivity by Suppressing RAD51 Expression and Homologous Recombination

**DOI:** 10.3390/ijms232112876

**Published:** 2022-10-25

**Authors:** Xile Pei, Emil Mladenov, Aashish Soni, Fanghua Li, Martin Stuschke, George Iliakis

**Affiliations:** 1Division of Experimental Radiation Biology, Department of Radiation Therapy, University Hospital Essen, University of Duisburg-Essen, 45147 Essen, Germany; 2Institute of Medical Radiation Biology, University Hospital Essen, University of Duisburg-Essen, 45147 Essen, Germany; 3German Cancer Consortium (DKTK), Partner Site University Hospital Essen, 45147 Essen, Germany

**Keywords:** PTEN, DSBs, DDR, HR, c-NHEJ, alt-EJ, SSA, PARP inhibitors, ionizing radiation

## Abstract

PTEN has been implicated in the repair of DNA double-strand breaks (DSBs), particularly through homologous recombination (HR). However, other data fail to demonstrate a direct role of PTEN in DSB repair. Therefore, here, we report experiments designed to further investigate the role of PTEN in DSB repair. We emphasize the consequences of PTEN loss in the engagement of the four DSB repair pathways—classical non-homologous end-joining (c-NHEJ), HR, alternative end-joining (alt-EJ) and single strand annealing (SSA)—and analyze the resulting dynamic changes in their utilization. We quantitate the effect of PTEN knockdown on cell radiosensitivity to killing, as well as checkpoint responses in normal and tumor cell lines. We find that disruption of PTEN sensitizes cells to ionizing radiation (IR). This radiosensitization is associated with a reduction in RAD51 expression that compromises HR and causes a marked increase in SSA engagement, an error-prone DSB repair pathway, while alt-EJ and c-NHEJ remain unchanged after PTEN knockdown. The G_2_-checkpoint is partially suppressed after PTEN knockdown, corroborating the associated HR suppression. Notably, PTEN deficiency radiosensitizes cells to PARP inhibitors, Olaparib and BMN673. The results show the crucial role of PTEN in DSB repair and show a molecular link between PTEN and HR through the regulation of RAD51 expression. The expected benefit from combination treatment with Olaparib or BMN673 and IR shows that PTEN status may also be useful for patient stratification in clinical treatment protocols combining IR with PARP inhibitors.

## 1. Introduction

The tumor suppressor gene *PTEN* (phosphatase and tensin homologue deleted on chromosome 10) was independently identified in 1997 by two groups while mapping homozygous deletions on human chromosome region 10q23, which is deleted or mutated in a considerable proportion of human cancers [1,2]. Germline mutations of *PTEN* are associated with hereditary cancer predisposition syndromes: Cowden syndrome (CS) and Bannayan-Zonana syndrome (BRRS) [3,4]. Mutations are inherited in an autosomal dominant manner and are characterized by multiple noncancerous, tumor-like growths called hamartomas and an increased risk of other malignancies [3,5,6]. The protein encoded by the PTEN gene is a phosphatidylinositol-3,4,5-triphosphate-3-phosphatase that shares sequence homology to the protein tyrosine phosphatase (PTPase) superfamily and with tensin, a cytoskeletal protein that links integrins to the actin cytoskeleton at sites of adhesion [7,8].

The protein encoded by the *PTEN* gene acts as a dual-specificity lipid phosphatase that functions as a direct antagonist of phosphoinositide 3-kinase (PI3K), which specifically catalyzes the dephosphorylation of the 3′ phosphate of the inositol ring in phosphatidylinositol (3, 4, 5)-trisphosphate (PIP3), resulting in the production of phosphatidylinositol-4, 5 bisphosphate (PIP2) [9]. This dephosphorylation is a vital process because it results in inhibition of the AKT/PKB (Protein kinase B) signaling pathway [10], which plays an important role in regulating critical cellular behaviors, such as glucose metabolism, cell growth, proliferation, survival and migration, through multiple downstream effectors [11]. The function of PTEN as an “off” switch for the AKT/PKB pathway explains its role as a strong tumor suppressor [9,10].

Hence, the decades after PTEN identification have witnessed the generation of an impressive body of experimental data supporting this notion. Somatic mutations or loss of PTEN can be found in numerous sporadic cancers, including endometrial, prostate, glioblastoma, thyroid, gastric, melanoma, pineal brain and small-cell lung, and are especially frequent in high degree malignancies [6,12,13,14,15]. Thus, PTEN loss or decreased expression have been recognized as a hallmark of malignant diseases and a strong negative predictor of patient survival after cancer treatment.

More recently accumulated evidence indicates that PTEN is involved in DNA damage repair [16]. Thus, the depletion of PTEN sensitizes tumor cells to radiotherapy that relies on the induction of DNA damage, particularly DSBs [17]. There is also a report that the disruption of PTEN is associated with reduction in RAD51 expression, a vital homologous recombination (HR) factor [18]. This implies that PTEN may serve in the clinic as a biomarker to identify HR-deficient tumors that can then be treated with poly-ADP ribose polymerase inhibitors (PARPi), ionizing radiation (IR) or cis-platinum (cis-DDP) [19]. However, other research groups failed to confirm these findings and to observe an association between PTEN loss and reduced RAD51 expression, even using the same cell lines as the original publication [19,20]. Some researchers disfavor the hypothesis that PTEN status affects RAD51 expression or its recruitment to DSBs [21] but agree that PTEN loss is associated with increased sensitivity to PARPi and certain DNA damaging agents [22,23,24]. Indeed, rather complex patterns of sensitivities to DNA damaging agents have been associated with the disruption of PTEN [19].

Despite their ground-breaking character, the above-presented studies leave open several questions regarding PTEN function in DNA damage response (DDR) in general and DSB repair in particular. In the present study, we focus on DSB repair and further explore the role of PTEN in repair pathway balance and cell cycle control after IR exposure. We reason that by delineating the role of PTEN in DSB repair, we will inform the next generation of PTEN effects on DSB repair in glioblastoma cell lines. Our results have the potential to improve customized therapeutic strategies of glioblastoma (GBM), a form of cancer with a high incidence of PTEN mutation/loss (~40%) [25].

Our results reveal an altered balance between DSB repair pathways after PTEN knockdown that is expected to also occur in tumors that lose PTEN function. Specifically, PTEN knockdown impairs HR, likely by downregulating RAD51 expression. HR suppression is compensated by single-strand annealing (SSA) that increases nearly twofold, while alternative end-joining (alt-EJ) and classical non-homologous end-joining (c-NHEJ) remain unaffected. The suppression of HR is accompanied by the expected suppression of the G_2_-checkpoint and by an increase in sensitivity to PAPRi, such as olaparib or BMN673. The benefit of a combination-treatment with olaparib or BMN673 and IR in PTEN negative tumors may, thus, be achievable.

## 2. Results

### 2.1. PTEN Knockdown Radiosensitizes Normal Epithelial RPE-1 and Glioblastoma M059K Cells

To study the functions of PTEN in DSB repair, we selected a human hTert–immortalized primary retinal epithelial cell line, RPE-1 (RPE-1 hTert), and a human glioblastoma cell line, M059K. Both cell lines retain PTEN function and express the protein at similar levels (Figure 1A). We screened three siRNAs in RPE-1 cells (siRNA-A, siRNA-B and siRNA-C; sequences under Materials and Methods) for knockdown efficiency. Western blot results of the first experiment show a similar degree of protein knockdown after transfection of RPE-1 hTert cells with siRNA-A or siRNA-B and no effect when siRNA-C was utilized (Appendix A). The second experiment confirms the equal effectiveness of siRNA-A and siRNA-B and forms the foundation of our selection of siRNA-B—henceforth simply siPTEN (Appendix A).

The transfection of RPE-1 or M059K cells with siRNA targeting PTEN causes 48 h later a profound reduction in PTEN expression (Figure 1A). We opted to knockdown PTEN and study its role on DDR and DSB repair pathway balance, rather than compare the responses of many different cell lines expressing or not the protein, because in this way we operate for each cell line in an isogenic system. In addition, we also minimize the influence of compensatory genetic changes caused by PTEN loss during the history of tumor development that can profoundly change the genetic background of the cells and influence thus the effects under investigation. Furthermore, the knockdown avoids the need for PTEN overexpression, frequently used with null cell lines, to generate PTEN proficient cells for comparison, as this is also associated with secondary effects.

Cell cycle analysis by flow cytometry shows that the distribution of cells throughout the cell cycle during the exponential phase of growth is similar in RPE-1 hTert and M059K cells and that PTEN knockdown fails to detectably change this distribution (Figure 1B,C). The analysis of radiosensitivity in RPE-1 hTert and M059K cells after PTEN knockdown is summarized in Figure 1D and Appendix A. It is evident that the depletion of PTEN causes marked increases of similar magnitude in cell radiosensitivity to killing, despite the fact that RPE-1 hTert cells are intrinsically more radiosensitive. Moreover, this conclusion is further supported by the survival results of U-2 OS osteosarcoma cell line after PTEN knockdown (Appendix A). Collectively, these data indicate that the depletion of PTEN can, in principle, radiosensitize both normal and tumor cells. More cell lines will need to be tested to confirm the generality of this effect and define the parameters that determine its magnitude.

### 2.2. The Role of PTEN in HR, SSA, c-NHEJ and alt-EJ

The observed radiosensitization of RPE-1 hTert and M059K cells after PTEN knockdown suggests effects on DSB repair. To screen how PTEN knockdown affects the function of the four available DSB repair pathways, we introduced a battery of four U-2 OS, GFP-reporter cell lines, a widely accepted standard in the field for such inquiries [26]. Each of these four cell lines, has stably integrated in their genomes a construct, that reports the function of a particular DSB repair pathway after processing a DSB introduced by the I-*SceI* meganuclease that is transiently expressed by transfecting the corresponding plasmid. Specifically, DR-GFP-U-2 OS cells report HR function, the SA-GFP-U-2 OS, SSA, EJ5-GFP-U-2 OS, NHEJ and EJ2-GFP-U-2 OS, alt-EJ [26].

Several reports implicate PTEN in HR [17,27]. We, therefore, firstly investigated how PTEN knockdown affects HR in U-2 OS DR-GFP cells (Figure 2A). The results show that effective PTEN knockdown causes non-detectable disturbances in cell cycle distribution (Appendix A). PTEN knockdown is in all U-2 OS reporter cell lines, similar to that in RPE-1 hTert and M059K cells (Appendix A). This correlates with the similar level of radiosensitization observed with U-2 OS cells (Appendix A).

PTEN knockdown reduces HR by about ~30%, as compared to cells transfected with a control siRNA. Because the effect on HR is modest we sought additional evidence to corroborate the outcome. It has been shown that the suppression of HR is often compensated by an increase in SSA. We, therefore, tested in a similar experiment the effect of PTEN knockdown in U-2 OS, SA-GFP cells. Notably, PTEN knockdown causes an increase in SSA by about 70% in these cells (Figure 2B), confirming compensatory actions following HR suppression. This result is significant for two reasons: first, it indirectly confirms the suppression for HR and second it shows that suppression of error-free HR is associated with a substantial increase in an error-prone DSB repair pathway, SSA. Similar experiments using U-2 OS, EJ2-GFP and U-2 OS, EJ5-GFP, reporting for alt-EJ and NHEJ, respectively, show that efficient PTEN knockdown (Appendix A) fails to generate detectable effects on these DSB repair pathways (Figure 2C,D).

### 2.3. Effect of PTEN on RAD51 Foci Formation

To further analyze the above observed reduction in HR after PTEN knockdown, we measured RAD51 foci formation and decay, specifically in cells irradiated in the G_2_-phase of the cell cycle that are proficient for HR. The results summarized in Figure 3 show that in both RPE-1 hTert and M059K cells exposed to 2 Gy of IR, RAD51 rapidly accrues to DSBs (Figure 3A,C) and forms foci that reach a maximum at 2 h at about 12 foci per cell that decay at later times (Figure 3B,D). The pronounced accretion of RAD51 to DSBs is reduced by almost 70% after PTEN knockdown, corroborating the observations with reporter cell lines that it suppresses HR.

Since HR is regulated at the level of DNA end resection, an initial step of HR, we measured the accumulation of RPA70 at single-stranded DNA generated during that process, by using flow cytometry in RPE-1 hTert and M059K cells exposed to 5 or 10 Gy (Figure 4A–D and Appendix A). Notably, the PTEN knockdown fails to detectably modify the level of DNA end resection in either cell line. 

We also analyzed resection at lower IR doses by measuring RPA70 foci formation using the same methodology employed to analyze RAD51 foci formation in G_2_-phase cells. The results shown in Figure 4E,F demonstrate effective resection detected by foci analysis in cells exposed to 2 Gy. There is extensive increase in RPA70 foci as a function of time, with a maximum reached at ~2 h, in line with the results on RAD51 accretion, both in RPE-1 hTert and M059K cells. However, even at the lower radiation doses tested with this assay, PTEN knockdown fails to modulate resection. We conclude that PTEN knockdown leaves resection unchanged and that the above observed reduction in HR should be attributed to other effects. In fact, the unchanged resection observed after PTEN knockdown explains the persistence of the SSA function shown above with the reporter cell line.

### 2.4. PTEN Knockdown Suppresses RAD51 Expression

To gain insights into the mechanism of HR suppression, we evaluated the levels of RAD51 protein in RPE-1 hTert and M059K cells after PTEN knockdown, specifically for cells enriched in the G_2_-phase of the cell cycle. Previous work suggests a PTEN-dependent regulation of the RAD51 expression [28]. The results in Figure 5A,B demonstrate that the RAD51 levels are markedly reduced after PTEN knockdown. This strong reduction in RAD51 levels is in line with the observed reduction in HR, but it remains open whether it is entirely and exclusively responsible for the effect observed.

There are reports that PTEN inhibition causes AKT hyperactivation, which, in turn, impairs HR - as we actually observe [27]. There are also reports for non-canonical, nuclear PTEN functions on DSB repair, some of which may compromise HR directly [17,29]. We, therefore, explore here the extent to which the above observed effects of PTEN knockdown derive from AKT hyperactivation. For this purpose, we treated RPE-1 hTert (Figure 5C and Appendix A), M059K (Figure 5D and Appendix A) and U2OS (Figure 5E and Appendix A) cells with bpV(HOpic) or SF1670, two specific PTEN inhibitors [30,31], at the indicated concentrations and treatment times and measured AKT activity, by analyzing the phosphorylated form of AKT at Serine 473 (AKT-pS473), a widely accepted proxy for activated AKT protein. In RPE-1 hTert cells (Figure 5C and Appendix A), treatment with either inhibitor causes AKT hyperactivation that is clearly detectable at 2 h and which increases with increasing treatment time and inhibitor concentration, being highest at the highest concentration used (100 nM).

Densitometry data shown on Appendix A help to visualize, in a quantitative manner, the activation of AKT. Qualitatively similar results are also obtained with M059K cells (Figure 5D and Appendix A). However, the densitometry results indicate that the SF1670 inhibitors has a higher impact on AKT phosphorylation in M059K cells (Appendix A). The clearest activation of AKT following PTEN inhibition is observed with U-2 OS cells (Figure 5E and Appendix A). Notably, despite clear AKT hyperactivation after treatment with PTEN inhibitors, the repair pathway reporter U-2 OS cell lines show no detectable effects on HR or SSA (Figure 6A). Additionally, survival assays carried out as a proxy for effects on HR activity show no radiosensitization after treatment with PTEN inhibitors, either in RPE-1 hTert or in M059K cells (Figure 6B,C). Based on these results we suggest that PTEN regulates HR in an AKT-independent manner. Notably, it is evident from the results shown in Figure 5 that treatment of cells with PTEN inhibitors fails to change the levels of RAD51 at least for the treatment times and inhibitor concentrations used. This result further hints to the relevance of RAD51 suppression in the suppression of HR.

### 2.5. Effect of PTEN on the Regulation of G_2_-Checkpoint

Previously, we have reported direct connections between the ability of cells to repair DSB by HR and the activation of the G_2_-checkpoint [32]. Since PTEN knockdown compromises HR, we considered it likely that it will also compromise the checkpoint response. This would construe further evidence for HR suppression following PTEN knockdown. When RPE-1 hTert and M059K cells are irradiated in G_2_-phase, they activate a checkpoint that delays their progression into mitosis and causes a rapid decrease in mitotic index (MI) that lasts for about 4 h, recovering at later times (Figure 7A,B and Appendix A). This response reflects the transient activation of the G_2_-checkpoint, specifically in G_2_-phase irradiated cells, as only these cells are able to reach mitosis during the time interval investigated [33]. Notably, in both RPE-1 hTert and M059K cells, the G_2_-checkpoint is markedly suppressed, i.e., the reduction in MI is lower after PTEN knockdown (Figure 7A,B). The activation of the G_2_-checkpoint in cells irradiated in S-phase can be analyzed in the same cell populations by following the accumulation of cells in G_2_-phase as a function of time after IR, using single parameter flow cytometry (Figure 7C,D).

Here, again a strong accumulation in G_2_-phase is observed starting 4 h after IR that is markedly reduced after PTEN knockdown. We conclude that PTEN knockdown suppresses HR and, as a consequence, the G_2_-checkpoint activation, both in cells irradiated in G_2_-phase, as well as in cells irradiated in S-phase.

### 2.6. PTEN Depletion Sensitizes Cells to Olaparib and BMN673

PARP inhibitors (PARPi) as single agents show promise in tumor treatment through the synthetic lethality they induce in cells with defects in BRCA1/BRCA2 and other components of the HR repair pathway [34,35]. The results outlined above show that PTEN knockdown is associated with radiosensitization and HR deficiency, which suggests sensitization to PAPR inhibitors, such as Olaparib or BMN673 [36,37]. To test this possibility, we conducted survival assays after administering Olaparib or BMN673 as monotherapy (Figure 8A).

Cells are plated at low density as required and 6 h later, after cells attach and recover from trypsinization, inhibitors are added at different concentrations and kept for 24 h. Growth medium is replaced thereafter and colonies are allowed to develop in the absence of inhibitors. The results in Figure 8A show that after PTEN knockdown, RPE-1 hTert and M059K cells are markedly more sensitive to treatment with Olaparib or BMN673, as expected from the associated HR defect. This effect is observed at all inhibitor doses tested. These results suggest that induced PTEN deficiency (here by knockdown) is a predictor of PARPi sensitivity, as already proposed in different settings [19,27].

We also inquired whether the depletion of PTEN enhances the radiosensitization usually observed after treating irradiated cells with PARPi. We carried out clonogenic survival assays combining IR with Olaparib (3 µM) or BMN673 (50 nM), given 1 h before and maintained for 5 h before plating. Figure 8B shows the results obtained. It is evident that when PTEN proficient cells are treated post-irradiation with Olaparib or BMN673, radiosensitivity is only marginally enhanced. Strikingly, following PTEN knockdown, administration of PARPi results in statistically significant radiosensitization of RPE-1 hTert and M059K cells, compared to PTEN knockdown-only cells. These results suggest a potentiation of the effect of PTEN deficiency on DSB repair by PARP inhibition, which may be a strategy in cancer therapy.

## 3. Discussion

### 3.1. PTEN Is a Tumor Suppressor with Functions in DSB Repair

Among the lesions induced in the DNA by diverse chemical or physical agents, the DSB is rather special because of the high risk for misrepair that is associated with its processing [38]. To counteract these risks, cells engage several pathways to remove DSBs from their genome: c-NHEJ, HR, alt-EJ and SSA. However, intriguingly, these multiple pathways are not equivalent alternatives, but show instead striking differences in speed and accuracy, as well as cell cycle dependence [39,40].

PTEN is a unique and bona fide tumor suppressor protein, which possesses both lipid and protein phosphatase activity and is inactivated in various human cancers [1,2,29]. Its inactivation causes AKT activation, which promotes cancer phenotypes and explains the role of PTEN as a strong tumor suppressor. Notably, however, there is also strong evidence for additional functions of PTEN that are related to its occasionally observed nuclear localization and which specifically affects genome stability and DSB repair [41,42]. In the last decade, the role of PTEN in DSB repair has been extensively studied, but several inconsistencies remain [19,21,27]. In the present study, we have demonstrated that PTEN is involved in the repair of DSBs via HR by regulating RAD51 expression. As a consequence, PTEN knockdown increases radiosensitivity, impedes G_2_-checkpoint and increases genetic instability. These results suggest that patients with PTEN loss might benefit from radiotherapy.

### 3.2. The Role of PTEN in DSB Repair Pathway Balance

We carried out clonogenic survival assays 48 h after efficiently ablating PTEN by siRNA. We found strong radiosensitization both in normal epithelial RPE-1 hTert cells, as well as in a malignant GBM cell line, M059K. The radiosensitization observed suggests an effect in the processing of IR induced DSBs, which, as noted above, are the main culprits of cell lethality under these conditions. From a detailed analysis of the balance of the four DSB repair pathways following PTEN knockdown, we were able to show, using U-2 OS GFP-reporter cell lines, that alt-EJ and NHEJ remain unchanged. Although one report shows that PTEN promotes c-NHEJ by regulating the expression of XLF [43], effects on c-NHEJ were not detectable in the cells used in the present study. On the other hand, HR decreased by over 30%. The decrease in HR is consistent with another recent report [27]. Notably, we also observed for the first time that PTEN knockdown not only suppresses HR but that it also increases SSA. Therefore, we propose that the suppression of the error-free HR repair pathway promotes error-prone processing by SSA, leading to the radiosensitization observed.

RAD51 is the key protein of HR that mediates homologous DNA pairing and strand exchange, a hallmark of HR [44]. Reduced RAD51 foci formation in irradiated cells deficient in PTEN confirms the HR defect, and the associated reduction in its expression provides a mechanistic explanation. 

A process that regulates DSB repair and controls DSB repair pathway choice is DNA end-resection, which consists of a 5′- to 3′-degradation of one strand on both sides of the DSB, creating two 3′-overhangs that are quickly protected by ssDNA-binding proteins, such as RPA [39,45]. It is well known that the length of resection and the cell cycle phase contribute to DSB repair pathway choice through mechanisms that are under intensive investigation. HR requires a 3′-ssDNA overhang at the DSB end to promote strand invasion into the sister chromatid, which contains the homology region used as template for DNA synthesis and repair [46]. We measured resection after PTEN knockdown by detecting RPA70, a subunit of the RPA heterotrimer, intensity utilizing flow cytometry and immunofluorescence. We observed similar RPA70 accretion as a function of time and IR-dose that was not affected by PTEN knockdown. Thus, the observed HR defect in PTEN deficient cells is unlikely to derive from a suppression of resection.

Furthermore, although PTEN inhibition causes the expected hyperactivation of AKT that has been implicated by itself in HR suppression, our results show that this line of HR regulation is unlikely to be the underpinning mechanism, as inhibition of PTEN function fails to recapitulate this effect or to cause radiosensitization. Therefore, we suggest that the contributions of PTEN to DSB repair are independent of its canonical function on the AKT signal transduction pathway [17]. 

There are reports that PTEN regulates *RAD51* transcription by regulating the *RAD51* promoter, or by regulating E2F1-mediated RAD51 expression [29,41]. However, this is not universally accepted [19,27]. The analysis of the effects of PTEN on RAD51 expression requires further investigations.

### 3.3. The Role of PTEN on the G_2_/M Checkpoint

Cells rely on cell cycle checkpoints to prevent cell division in the presence of DSBs that lead to chromosome breakage. These checkpoints are rapid signaling responses that delay the progression of cells through the cell cycle, change transcription and mobilize the DNA repair machinery [47]. Among checkpoints, the G_2_-checkpoint is particularly relevant, as it prevents cells from entering into M-phase with unpaired DSBs.

In the present study, we investigated the function of PTEN in the G_2_-checkpoint and analyzed its function separately for cells irradiated in G_2_- or S-phase. The results show that PTEN knockdown causes a partial suppression of this checkpoint for cells irradiated in either phase of the cell cycle. We consider this effect as secondary to HR suppression, rather than as evidence for the direct involvement of PTEN in this cellular response to IR, as we have reported that HR deficiencies uniformly suppress G_2_-checkpoint activation [32].

### 3.4. PTEN Deficient Cells Are Sensitized to Olaparib or BMN673 with or without IR

PARPis have been approved by the U.S. Food and Drug Administration (FDA) and the European Medicines Agency (EMA) for the treatment of several cancers with BRCA1/BRCA2 mutations, or other deficiencies in HR components. As a consequence, there are ongoing phase 2 and 3 clinical trials for expanding their applications in cancer treatment [48]. In the present study, we show that PTEN deficient cells not only are sensitized to PARPis, as already reported [19,27], but are also further sensitized to IR [19,27]. This suggests a promising potential for their application in radiotherapy of PTEN-deficient tumors. However, more investigations are required in order to generate a reasonable rationale for combined treatment of PTEN-deficient tumors.

The present study recapitulates our interest in the comprehensive treatment of GBM. GBM is a highly malignant brain tumor with a limited median survival ranging from 12 to 16 months after diagnosis [49]. Apart from surgery, radiotherapy and chemotherapy (TMZ used predominantly), novel and more efficient treatment modalities are urgently needed. The results of this study provide foundations for broadening our treatment horizons for GBM treatment, especially for patients with PTEN loss or mutation.

However, as with other targeted therapies, resistance to PARPi arises in advanced disease. Because GBM is a malignant tumor with high heterogeneity, there must be some other elements or genetic characteristics in addition to PTEN, which might affect the efficacy of PARPi. One of them might be alterations in DSB repair pathway balance. So, determining the optimal use of PARPi within combination treatments that include IR will be challenging but worth investigating.

In conclusion, PTEN is involved in the regulation of DSB repair via HR. The depletion of PTEN increases cell radiosensitivity to killing by downregulating RAD51 expression through mechanisms that remain to be elucidated. The loss of PTEN function also impedes the G_2_-checkpoint and increases genomic instability. Notably, cells deficient in PTEN benefit from a combination treatment with IR and olaparib or BMN673, which may assist the design of individualized strategies for GBM therapy, taking advantage of the highly localized application of IR that reduces effects on normal tissues.

## 4. Materials and Methods

### 4.1. Cell Lines, X-ray Irradiation and Drug Treatments

RPE-1 hTert and M059K cells were cultured as monolayers in cell culture dishes with D-MEM growth medium and supplemented with 10% fetal bovine serum (FBS), 100 μg/mL penicillin and 100 μg/mL streptomycin.

U-2 OS DR-GFP, U-2 OS SA-GFP, U-2 OS EJ2-GFP and U-2 OS EJ5-GFP were grown in McCoy’s 5A growth medium, supplemented with 10% FBS, 100 μg/mL penicillin and 100 μg/mL streptomycin and 2 µg/mL puromycin. All cell lines were cultivated at 37 °C in a humidified incubator with 5% CO_2_. All cell lines were routinely checked for mycoplasma contamination and only mycoplasma free cells were used in the corresponding experiments.

Cells were irradiated at room temperature using an X-ray generator (GE Healthcare) operated at 320 kV, 10 mA with a 1.65 mm aluminum filter (effective photon energy ∼70–90 kV). To achieve an optimal dose distribution, the distance between X-ray tube and irradiation table was adjusted according to the dish size. The 35 mm and 60 mm diameter cell-culture dishes and 25 cm^2^ flasks were irradiated at a distance of 50 cm, while 100 mm diameter dishes and 75 cm^2^ flasks were irradiated at 75 cm distance from radiation source. The dose rates at 50 cm and 75 cm distance were ~3.6 Gy/min and ~1.6 Gy/min, respectively. To avoid temperature fluctuations while performing the G_2_/M checkpoint assay, cells were irradiated on a warm plate at 37 °C. Controls were treated identically but were not exposed to radiation. 

Olaparib and BMN673 were administrated to the cells 1 h before IR at concentrations of 3 µM and 50 nM, respectively. Cells were plated 5 h after irradiation for further analysis. PTEN inhibitors (bpV(HOpic) and SF1670) were added into the growth media 2 h before IR at the indicated concentrations. Cells were incubated for 6 h after irradiation and then plated for colony formation. Control cells were treated with the corresponding concentrations of DMSO. In the clonogenic survival assays, the growth medium, containing the inhibitors, was replaced 24 h later using inhibitor-free growth medium and cells were allowed to from colonies.

### 4.2. RNA Interference

Transfection of siRNA into cells was performed using the GenePulser X cell electroporation apparatus (Bio-Rad). The siRNAs were as follows: PTEN-siRNAs, siRNA-A: 5′-CACACAGCUAGAACUUAU-3′, siRNA-B: 5′-CCAGUCAGAGGCGCUAUGU-3′, [50,51,52], and siRNA-C: 5′-AGUGGCGGAACUUGCAAUC-3′, negative control (siNC): 5′-UUCUCCGAACGUGUCACG U-3′. The specificity and efficiency of all 3 PTEN-siRNAs, were determined by Western blot analysis of PTEN level after transfection with specific PTEN siRNA (Appendix A). Due to its best performance, in all further experiments, siRNA-B, simply indicated as siPTEN, was utilized. For transfection, cells were harvested by trypsinization, centrifuged at 900 rpm for 3 min and resuspended at 4 × 10^6^ cells per 100 μL of HB buffer. Cells were transferred into an Ingenio^®^ cuvette (Mirus, Medison, WI, USA) and electroporated with the program achieving optimal transfection efficiency for the corresponding cell line. Transfection efficiency was estimated by transfecting cells with a GFP expressing plasmid. The number of GFP positive cells detected by flow cytometry 24 h after transfection served as a measurement of transfection efficiency. Controls were transfected with a negative-control siRNA.

### 4.3. Clonogenic Survival Assays

Clonogenic survival assays were employed to estimate cell radiosensitivity to killing. After transfection with control siRNA or PTEN siRNA, cells were seeded in 60 mm dishes and incubated at a density of approximately 10^5^ cells/dish for the control group and 2 × 10^5^ cells/dish for the PTEN knockdown group for 48 h. Subsequently, cells were irradiated and seeded at low numbers in 60 mm dishes, aiming 30 to 150 colonies/dish. After 11 days of growth for RPE-1 hTert cells and 15 days for M059K cells, colonies were stained with 1% crystal violet in 70% ethanol, counted under the microscope and used to calculate plating efficiency and cell survival.

### 4.4. Flow Cytometry (FC) Analyses of Cell Cycle Distribution, Mitotic Index (H3-pS10) and DNA End-Resection

Cell cycle distribution was measured by FC analysis of DNA content after staining with propidium iodide (PI). Cells were collected and fixed overnight in ice-cold 70% ethanol and were subsequently resuspended in PI staining solution (40 μg/mL PI and 62 μg/mL RNaseA in PBS) for 15 min at 37 °C. Analysis was carried out with a Gallios^®^ flow cytometer (Beckman-Coulter, Krefeld, Germany).

Two-parameter flow cytometry analyses were employed for determination of the mitotic index (MI). For this experiment, cells were plated in 25 cm^2^ cell culture flasks and were grown for 24 h. Flasks were then transferred to a warm room at 37 °C for the remaining manipulations. At the indicated time points, cells were collected and fixed overnight in ice-cold 70% ethanol. Subsequently, cells were spun-down and cell pellets incubated for 15 min in 500 μL permeabilization solution (ice-cold PBS + 0.25% Triton X-100) and collected by centrifugation at 1500 rpm. Cells were blocked in 500 μL PBG buffer (0.2% gelatin, 0.5% bovine serum albumin in PBS) for 1 h at RT. Cells were incubated for 2 h at RT in 150 µL primary antibody solution (anti-H3-pS10, mouse-monoclonal, Abcam, Cambridge, UK), diluted at 1:2000 in PBG. After two washing steps with PBS, cells were incubated for 1.5 h at RT in 150 µL secondary antibody solution, (anti-mouse Ab, conjugated with AlexaFluor^®^488, Life Technologies, Taufkirchen, Germany). Cell pellets were incubated with PI staining solution for 15 min at 37 °C and were analyzed on a Gallios^®^ flow cytometer. At least 30,000 events were scored. The acquired data was further analyzed and visualized by Kaluza software (Beckman-Coulter, Krefeld, Germany).

For three parametric analysis of DNA end resection, cells were plated in 100 mm dishes and grown for 48 h. Cells were pulse-labeled with 10 µM EdU for 30 min before exposure to IR. After exposure to IR, cells were collected at different time points and permeabilized in 500 μL of ice-cold 1 × PBS containing 0.25% Triton X-100 for 2 min on ice. Cells were fixed in 500 μL 3% PFA solution containing 2% sucrose for 15 min at RT. Additionally, they were blocked overnight by incubation in 500 µL PBG.

To quantify DNA end-resection, an anti-RPA70 antibody was utilized. Cells were incubated for 1.5 h in 100 μL primary antibody solution (anti-RPA70 monoclonal antibody), purified in-house (αSSB70B, mouse hybridoma cell line kindly provided by Dr J. Hurwitz). Cells were washed twice in PBS and incubated in 100 μL of secondary antibody (anti-mouse, conjugated with AlexaFluor^®^488, Life Technologies, Taufkirchen, Germany) for 1 h at RT. EdU incorporation was visualized by incubation of samples in 100 μL of EdU ClickiT reaction for 30 min at RT. Cells were washed once more with PBS and DNA was stained with PI solution for 15 min at 37 °C. Samples were measured with a Gallios^®^ flow cytometer. Only G_2_-phase, EdU-negative cells were analyzed for resection, as described earlier [26].

### 4.5. Indirect Immunofluorescence Staining and Quantitative Image-Based Cytometry (QIBC) Foci Analysis

For indirect immunofluorescence detection of IR-induced foci (IRIF), cells were plated on glass coverslips placed in 35 mm cell culture dishes. Before IR, cells were pulse-labeled with 10 µM EdU for 30 min, then medium containing EdU was exchanged with pre-warmed fresh medium. At the corresponding time intervals after IR, cells were fixed in 3% PFA; 2% sucrose solution for 15 min and were permeabilized in P-solution (100 mM Tris pH 7.4, 50 mM EDTA pH 8.0, 0.5% Triton X-100) for 10 min at RT. Subsequently, cells were washed with PBS and blocked in PBG blocking buffer (0.2% gelatin, 0.5% bovine serum albumin in PBS) overnight at 4 °C. For detection of RPA70 foci, an initial permeabilization step with 0.25% Triton X-100 in PBS was performed.

For detection of RAD51 or RPA70 foci, cells were incubated with the primary antibodies diluted in PBG; the primary antibodies used were: anti-RPA70 (αSSB70B) and anti-RAD51 (mouse monoclonal, Gene Tex, Irvine, CA, USA). Cells were incubated with primary antibodies for 1.5 h at RT, and after three washing steps were incubated with AlexaFluor^®^-conjugated secondary antibodies (anti-mouse, AlexaFluor^®^647, Abcam, Berlin, Germany). The EdU ClickiT (Thermo Scientific, Bremen, Germany) reaction was performed according to the manufacturer’s instructions. Cells were counterstained with 200 ng/mL 4′, 6-diamidino-2-phenylindole (DAPI) dissolved in PBS for 5–10 min at RT in order to visualize nuclei. Cells were additionally washed in PBS and coverslips were mounted on microscope slides using the PromoFluor Antifade Reagent (PromoKine, Heidelberg, Germany). Images were captured by AxioScan Z1, automated platform (ZEISS, Jena, Germany) and foci analyses were carried out by Imaris^®^ software (Bitplane, Oxford, UK).

### 4.6. SDS-PAGE and Immunoblotting

Proteins from whole cell lysates were extracted using radioimmunoprecipitation assay (RIPA) buffer supplemented with protease and phosphatase inhibitor cocktails, according to the manufacturer’s instructions (Themo Scientific, Germany). The 2 × 10^6^ cells were lysed in 50–100 μL of RIPA buffer. Protein concentrations were determined by the Bradford assay and 50 μg of total protein extracts were mixed with 2 × Laemmli sample buffer.

Resolved proteins were transferred to a nitrocellulose membrane by wet transfer and were blocked with 5% non-fat dry milk in TBS, supplemented with 0.5% Tween 20 (TBS-T). Primary antibodies were diluted in 5% non-fat dry milk in TBS-T and incubated with the membrane overnight at 4 °C. After three times washing with TBS-T, the membrane was incubated for 1 h at RT with secondary antibodies diluted in TBS-T. The following primary antibodies were used: anti-PTEN, mouse monoclonal (Santa Cruz, Heidelber, Germany), anti-RAD51, mouse monoclonal (Gene Tex, Irvine, CA, USA), anti-β-Actin, rabbit polyclonal (Gene Tex, Irvine, CA, USA), anti-AKT-pS473, mouse monoclonal (Santa Cruz, Heidelber, Germany). The secondary antibodies were: anti-rabbit IgG, conjugated with IRDye680 (Li-COR Biosciences, Bad Homburg, Germany) and anti-mouse IgG, conjugated with IRDye800 (Li-COR Biosciences, Bad Homburg, Germany). Prior to detection, the membrane was washed again, as described above, and was allowed to dry. For detecting proteins of interest, an Odyssey^®^ Infrared Imaging System (LI-COR Biosciences, Bad Homburg, Germany) was used. All RAW uncropped information about the Western blot membranes is provided as Appendix A).

### 4.7. GFP Reporter Cell Lines to Measure HR, SSA, alt-EJ, and NHEJ Activity at I-SceI Induced DSBs

U-2 OS GFP reporter cell lines (a gift of Dr J. Stark) [53] were exploited to measure the repair of I-*SceI* induced DSBs by a specific DSB repair pathway. The DR-GFP reporter cell line is specific for HR, the EJ5-GFP for NHEJ (total end-joining or distal end-joining), the SA-GFP for SSA and the EJ2-GFP for alt-EJ [26]. These reporters are stably integrated into the genomes of the corresponding cell lines. Transfection of cells with I-*SceI* expression plasmid (1 µg pDNA/10^6^ cells) was carried out 48 h after transfection with siRNA. The I-*SceI* mediated DSB was, thus, induced at the corresponding site and its repair by the indicated DSB repair pathway generated a GFP signal that was measured with a Gallios^®^ flow cytometer (Beckman Coulter) 24 h later. For the reporter assays, including bpV(HOpic) and SF1670 treatments, the drugs were administrated 6 h after transfection, when cells were completely adherent and kept until cells were collected for flow cytometry analysis.

### 4.8. Statistical Analysis

All statistical analysis was carried out by using an online version of MedCalc Software (MedCalc Software Ltd. Comparison of means calculator. https://www.medcalc.org/calc/comparison_of_means.php (Version 20.115; accessed on 5 October 2022)). The comparison of means module calculates the difference between the observed means in two independent samples. A significance value (*p*-value) is the probability of obtaining the observed difference between the samples if the null hypothesis were true. The comparison of means algorithm utilizes a two-tailed Student’s t-test to calculate the *p*-value.

The ANOVA analysis, implemented in Figure 6A was generated by the online applet (https://statpages.info/anova1sm.html, (accessed on 17 October 2022)), which also applies the Tukey HSD (“Honestly Significant Difference”) post hoc test, to indicate the significance between different groups. All the data, represents the means and standard deviations from three independent experiments. The detailed data of the statistical analysis is included in the corresponding Appendix A.xlsx files.

## Figures and Tables

**Figure 1 ijms-23-12876-f001:**
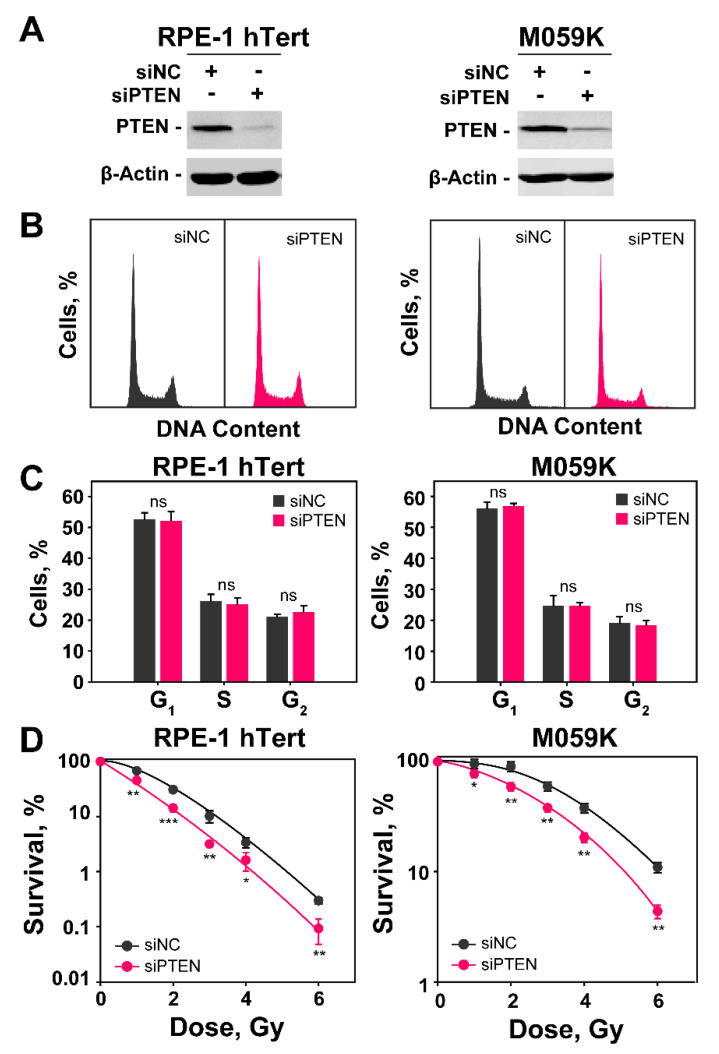
*PTEN knockdown radiosensitizes RPE-1 hTert and M059K cells.* (**A**) Western blot analysis of PTEN in RPE-1 hTert and M059K cells transfected with specific siRNA targeting PTEN protein; β-actin serves as loading control. (**B**) Flow cytometry histograms of RPE-1 hTert and M059K cells after transfection with PTEN siRNA. (**C**) Distribution of PTEN knock-down RPE-1 hTert and M059K cells in the different cell cycle phases. The analysis shows no significant differences between negative control (siNC) and PTEN knock-down (siPTEN) cells. (**D**) Clonogenic survival experiments of RPE-1 hTert and M059K cells transfected or not with PTEN siRNA. Data represent the mean ± SD from three independent experiments. The significance level, or *p*-value, is calculated using the two-tailed, Student’s *t*-test: ns (not significant), * *p* < 0.05, ** *p* < 0.01, *** *p* < 0.001.

**Figure 2 ijms-23-12876-f002:**
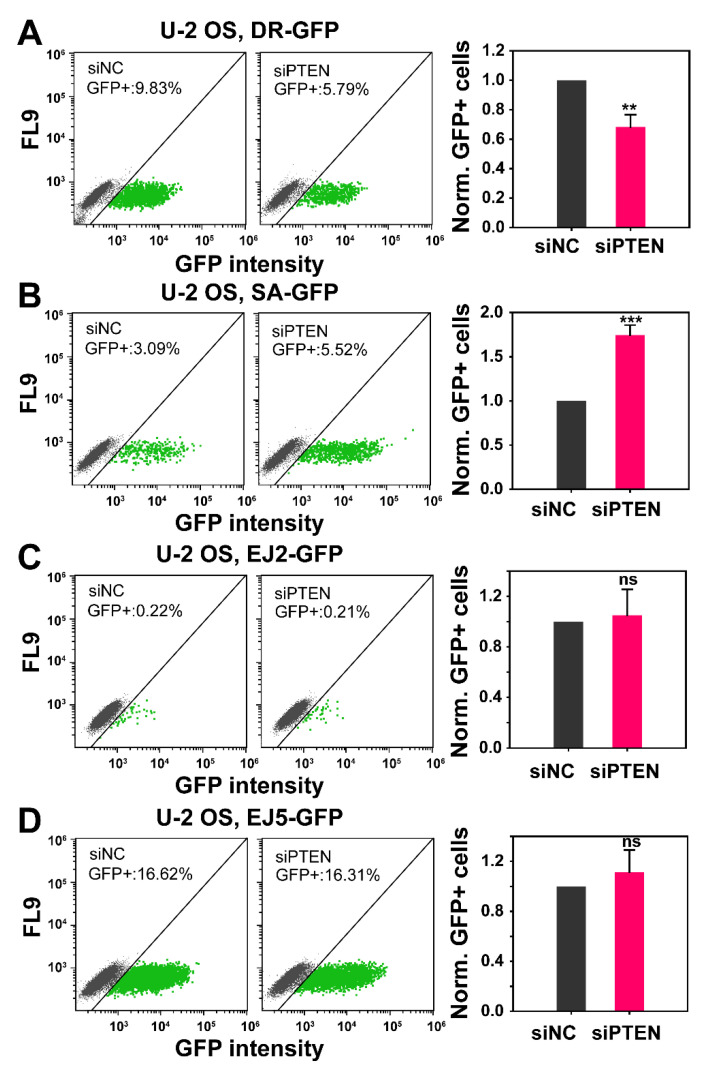
*Effect of PTEN deficiency on HR, SSA, NHEJ and alt-EJ.* The established GFP-reporter assays in U-2 OS cells, specifically designed to report the repair of I-*SceI*-induced DSBs by HR (DR-GFP), SSA (SA-GFP), NHEJ (EJ5-GFP) and alt-EJ (EJ2-GFP) were utilized. (**A**) Percentage of GFP positive cells (GFP+), in the negative control (siNC) and PTEN knock-down (siPTEN) of DR-GFP cells. Bar plots (right panel) reflect the siNC-normalized GFP+ cells. (**B**) Same as panel (**A**), but for SA-GFP cells. (**C**) Same as panel (**A**), but for EJ5-GFP cells. (**D**) Same as panel (**A**), but for EJ2-GFP cells. Data represent the mean ± SD from three independent experiments. The significance level, or *p*-value, is calculated using the two-tailed, Student´s *t*-test: ns (not significant), * *p* < 0.05, ** *p* < 0.01, *** *p* < 0.001.

**Figure 3 ijms-23-12876-f003:**
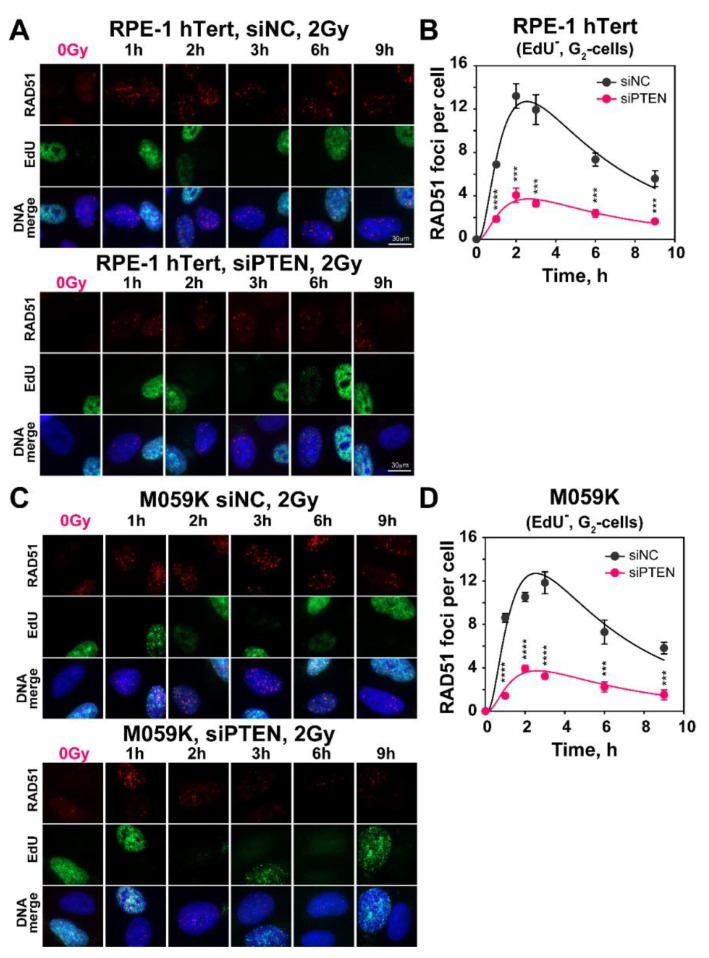
*PTEN deficiency results in decreased number of IR-induced RAD51 repair foci.* (**A**) Representative images of RAD51 foci in RPE-1 hTert cells, irradiated with 2 Gy of X-rays and collected at the indicated times after irradiation. Forty-eight hours before irradiation, cells were transfected with siNC or siPTEN, siRNA. (**B**) Quantification of RAD51 foci in siNC and siPTEN, RPE-1 hTert, EdU^−^, G_2_-cells. (**C**) Same as panel (**A**), but for M059K cells. (**D**) Same as panel (**B**), but for M059K cells. Cell cycle-specific RAD51 foci analysis was performed in EdU negative, G_2_-phase cells (EdU-, G_2_-cells), as described in Materials and Methods. The scale bar is 30 µm for all images. Data represent the mean and ± SD from three independent determinations. The significance level, or *p*-value, is calculated using the two-tailed Student´s *t*-test: ns (not significant), * *p* < 0.05, ** *p* < 0.01, *** *p* < 0.001, **** *p* < 0.0001.

**Figure 4 ijms-23-12876-f004:**
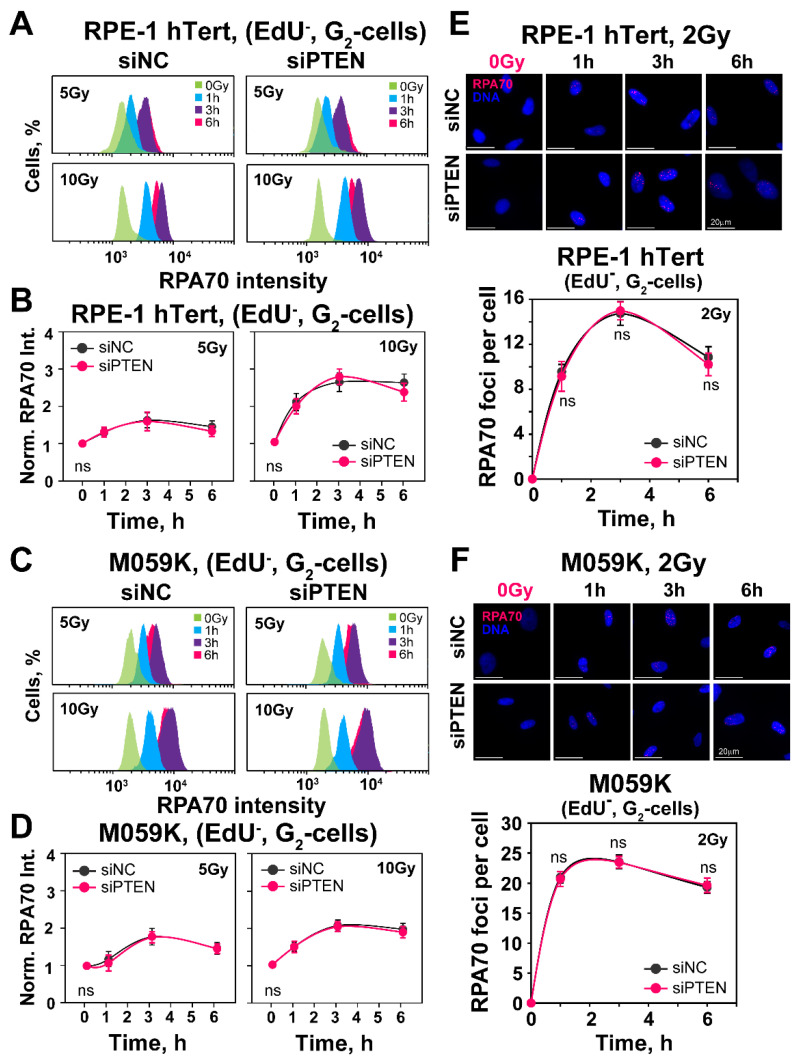
*PTEN knock-down fails to change DNA-end-resection in RPE-1 hTert or M059K cells.* (**A**) Representative flow cytometry histograms of RPA70 intensity signal in EdU-negative, G_2_-phase RPE-1 hTert cells (EdU^−^, G_2_-cells). (**B**) Quantification of the data shown in panel (**A**). The radiation effect on DNA end resection is shown after normalization of RPA70 signal, by dividing the mean signal intensity of irradiated cells by that of non-irradiated cells. (**C**) Same as panel (**A**), but for M059K cells. (**D**) Same as panel (**B**), but for M059K cells. (**E**) Representative IF images of RPA70 foci in exponentially growing RPE-1 hTert cells exposed to 2 Gy of X-rays. RPA70 foci are scored in EdU-negative, G_2_-phase cells (EdU-, G_2_-cells), as described in Materials and Methods. (**F**) Same as panel (**E**), but for M059K cells. The scale bar is 20 µm for all images. Data are means ± SD from three independent determinations. The significance level, or *p*-value, is calculated using the two-tailed Student´s *t*-test: ns (not significant), * *p* < 0.05.

**Figure 5 ijms-23-12876-f005:**
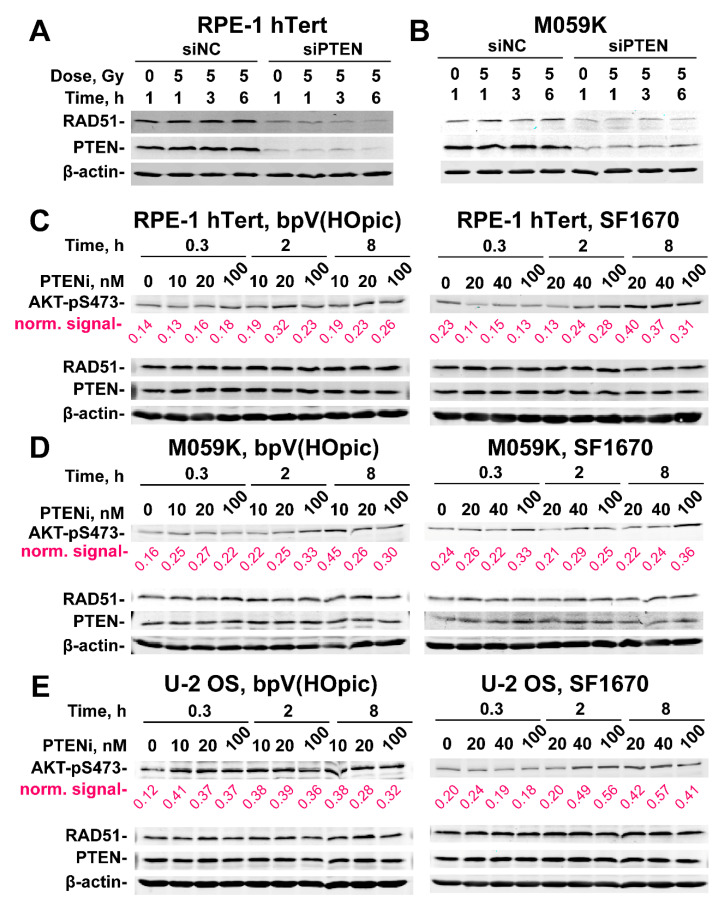
*PTEN knock-down suppresses RAD51 expression.* (**A**) Effect of PTEN deficiency on RAD51 expression in RPE-1 hTert cells. β-Actin serves as loading control. (**B**) Same as panel (**A**), but for M059K cells. (**C**) Western blot analysis of AKT-pS473, PTEN and RAD51 protein levels in RPE-1 hTert cells treated with the indicated concentrations of bpV(HOpic) or SF1670 (PTEN inhibitors) for the indicated times. (**D**) Same as panel (**C**), but for M059K cells. (**E**) Same as panel (**C**), but for U-2 OS cells. Densitometry analysis of the gels are shown in Appendix A. The β−actin normalized values obtained from densitometry analysis are indicated in magenta below each lane.

**Figure 6 ijms-23-12876-f006:**
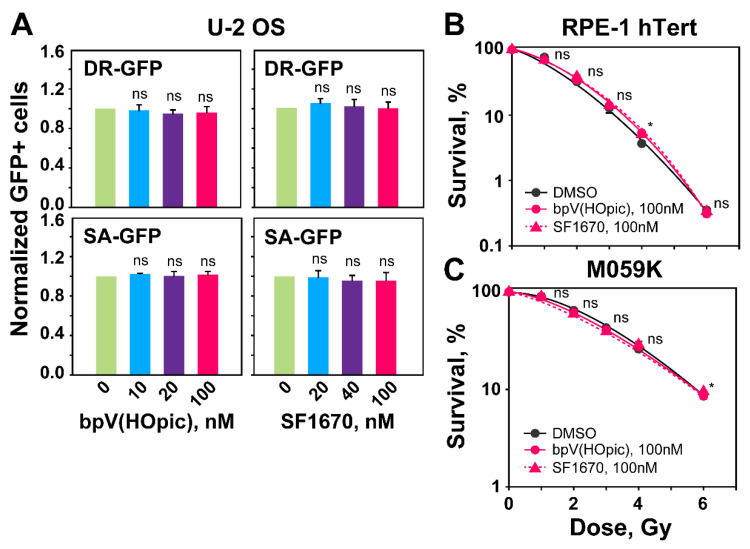
*PTEN inhibition by bpV(HOpic) or SF1670 leaves unaffected the levels of HR or SSA determined by GFP-reporter assays.* (**A**) DR-GFP and SA-GFP reporter U-2 OS cells treated with the indicated concentrations of PTEN inhibitors. GFP-positive cells (GFP+) are measured 48 h after transfection with I-SceI expression plasmid. (**B**) Clonogenic survival assays with RPE-1 hTert cells treated with the indicated concentrations of PTENi. (**C**) Same as panel (**B**), but for M059K cells. Data represent the means ± SD from three independent experiments. ANOVA analysis with Tukey HSD post hoc test is used to calculate the statistical significance for the data plotted in Figure 6A, while the significance level in Figure 6B,C is calculated using the two-tailed Student´s *t*-test: ns (not significant), * *p* < 0.05.

**Figure 7 ijms-23-12876-f007:**
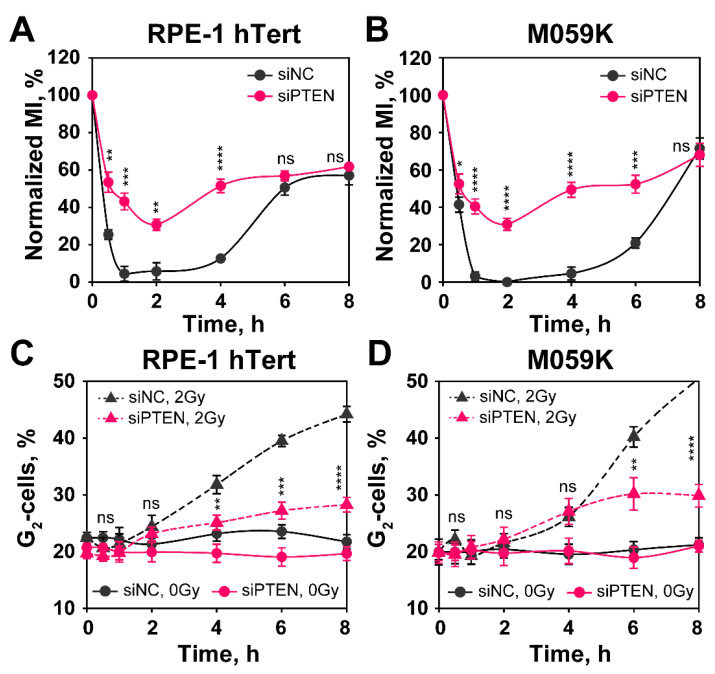
*PTEN knock-down impairs G_2_-checkpoint activation in RPE-1 hTert and M059K cells.* The checkpoint activated in cells that are in G_2_-phase at the time of irradiation is measured by analyzing the mitotic index (MI), using two-parameter flow cytometry detecting DNA through PI staining and phosphorylated H3 at Serine 10 (H3pS10), a specific marker of mitotic cells, by antibody staining. (**A**) Normalized MI in siNC and siPTEN transfected RPE-1 hTert cells. Normalized MI is calculated by dividing the MI measured in irradiated cells by that of non-irradiated cells. (**B**) Same as in panel (**A**), but for M059K cells. (**C**) Single parametric flow cytometry analysis of siNC and siPTEN transfected RPE-1 hTert cells showing the activation of the G_2_-checkpoint of S-phase irradiated cells. (**D**) Same as in panel (**C**), but for M059K cells. Data from three independent experiments are presented as mean ± SD. The significance level, or *p*-value, is calculated using the two-tailed Student´s *t*-test: ns (not significant), * *p* < 0.05, ** *p* < 0.01, *** *p* < 0.001, **** *p* < 0.0001.

**Figure 8 ijms-23-12876-f008:**
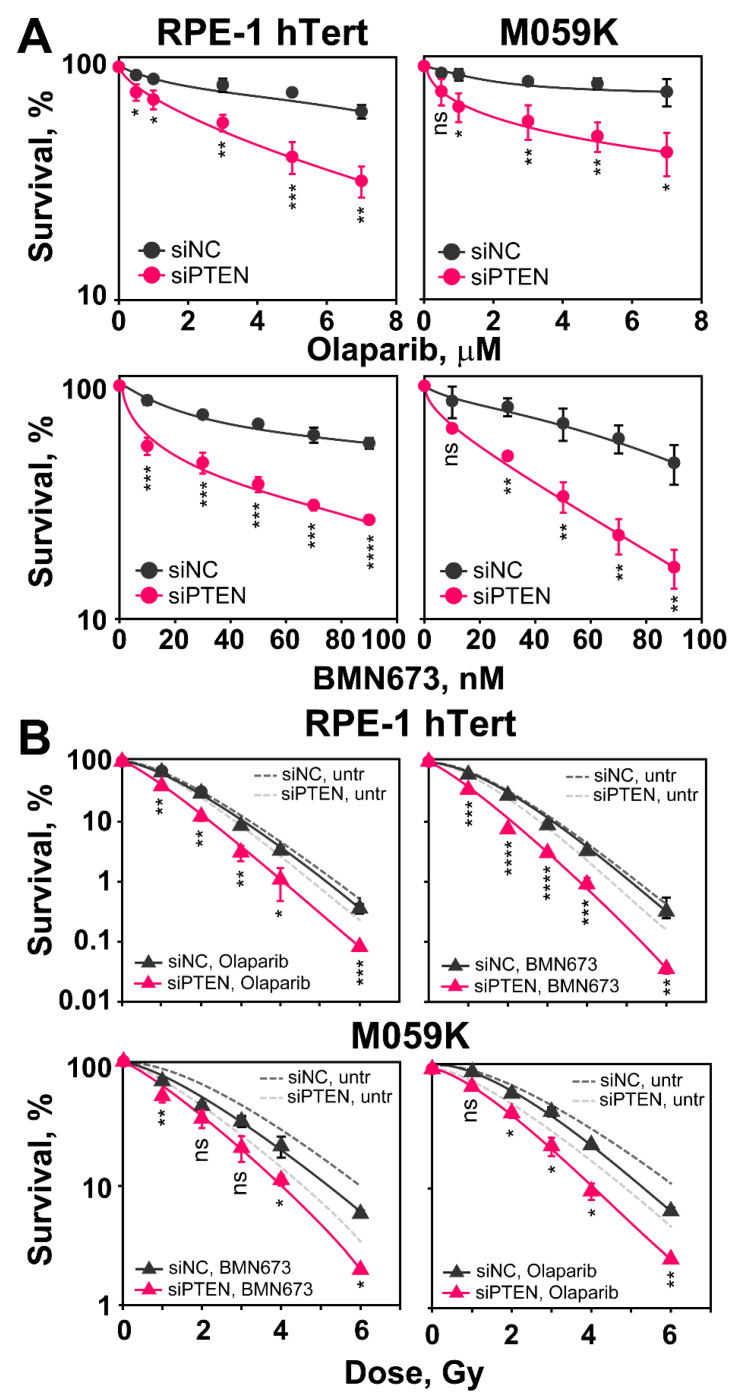
*PTEN knock-down suppresses HR and renders non-irradiated and irradiated cells sensitive to PARPi, Olaparib and BMN673*. (**A**) Effect of Olaparib and BMN673 on the survival of RPE-1 hTert and M059K cells. Cells are treated with the indicated PARPi concentrations for 24 h. (**B**) Clonogenic survival assays of RPE-1 hTert and M059K cells exposed to increasing doses of X-rays and treated with PARPi (Olaparib—3 μM or BMN673—50 nM). Data show the mean ± SD from three independent experiments. The significance level, or *p*-value, is calculated using the two-tailed Student´s *t*-test: ns (not significant), * *p* < 0.05, ** *p* < 0.01, *** *p* < 0.001, **** *p* < 0.0001. Note that the significance of the observed effect is cell line dependent.

## Data Availability

All included data will be available on request to the corresponding author.

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
