# Peer review of "PTEN Loss Enhances Error-Prone DSB Processing and Tumor Cell Radiosensitivity by Suppressing RAD51 Expression and Homologous Recombination"

_ijms, 2022, doi:10.3390/ijms232112876_

Round 1
Reviewer 1 Report
The DNA double-strand break (DSB) repair machinery is critical for maintaining genomic integrity and eliminating genotoxic stresses that could become oncogenic. However, by inducing DNA DSBs via chemotherapy or ionizing radiation in cells that are deficient in their ability to repair these DSBs, tumor cells can be treated and eliminated effectively, albeit for a short time in many cases. Therefore, understanding the molecular mechanisms that underlie how these cells can be made more sensitive to ionizing radiation or therapies targeting these DSB repair pathways will be critical in the development of novel therapeutic regimens. There are four main types of DNA DSB repair pathways – homologous recombination, single strand annealing (SSA), classical non-homologous end joining, and alternative end-joining. PTEN, one of the most altered tumor suppressors in human cancers, has been shown by other groups to be involved in DNA damage repair, with major attention given to homologous recombination. However, little is known about PTEN’s role in the other major DNA DSB repair pathways. Here, Pei et al show that by losing PTEN expression there is a decrease in homologous recombination repair of DNA DSBs with concomitant increases in activity of the SSA DNA damage repair pathway. No change in the two end-joining pathways was observed. The SSA DNA damage repair pathway is highly error prone. The induction of this pathway and the reduction of the homologous recombination pathway is postulated as the mechanism underlying the increased sensitivity of PTEN-depleted cells to DNA damage-inducing ionizing radiation and PARP inhibitors. This work is a step forward in understanding PTEN’s role in regulating DNA DSB repair across all four major DSB repair pathways in retinal epithelial cells, glioblastoma, and osteosarcoma cell lines. However, the impact of the work is dampened because of the use of cell lines whose connection to one another is unclear and deficits in the rigorous use of siRNAs, statistical reporting, and over-interpretation of the results for glioblastomas. See comments below:
Major comments:
1) Pei et al use an immortalized retinal epithelial (RPE-1 hTert), a glioblastoma (M059K), and an osteosarcoma (U2OS) cell line throughout the work. It is not clear how generalizable the results obtained in any one of the three model systems would be for the other cell lines used in this manuscript or for the cancer types that they represent. It is somewhat understandable why U2OS cells were used as they already expressed the DSB repair pathway constructs, it seems a more representative experiment would have been to include these constructs in either the RPE-1 hTert or M059K cell lines as well. Therefore, more impactful results would be obtained if 1) the constructs were added to RPE-1 or M059K and the U2OS results repeated in the newly generated cell lines and if 2) more cancer-specific cell lines – such as a second or third retinal epithelial, glioblastoma, and osteosarcoma cells – were also included. This would increase the generalization of the results, especially with the focus of the Introduction and the Discussion on glioblastoma cells. These experiments would need to be repeated throughout the manuscript with the additional cell lines.
2) A major drawback of the work is the use of only one siRNA targeting PTEN. This is unacceptable. Multiple siRNAs performed individually or pooled together should be used for these studies. By only using one siRNA in the study, you discount any effects that may be due to off-target effects by the siRNA.
3) The rationale for the use of RPE-1 hTert and M059K having relatively similar levels of PTEN expression is not entirely valid. Based on the information provided in the original immunoblots, these two sets of cell line experiments were not run on the same immunoblot. Even minor alterations in the buffer conditions (from the beginning of the immunoblot with SDS-PAGE running, transfer, blocking, antibody, or wash buffers) or chemiluminescent exposure times can cause inconclusive comparisons between blots. These protein lysates need to be run on the same immunoblot.
4) Although most of the Materials and Methods section is written well and in-depth, there are major issues with two aspects, detailed below.
a. There is no clearly defined Statistical Analysis section. There needs to be dedicated space to list all the statistical tests used in the paper. The only test noted in the manuscript was Student’s t test. That needs to be rectified. Furthermore, only the quantification of the flow cytometry plots in Figure 2 have p-values. A statistical test needs to be performed to display a p-value for the reader for multiple Figures (1C, 1D, 3B, 3D, 4B, 4D, 4E, 4F, 6-8 (all panels), and S1D).
b. The cell lines need to be authenticated, via methods such as STR analysis, to confirm that the cells that are used for the experiments are what they say they are. Furthermore, there is no mention of mycoplasma testing. Mycoplasma-positive cultures can produce erroneous results across a variety of experimental readouts. Testing for mycoplasma and verification that the cell lines are mycoplasma negative will be needed.
5) The conclusions from the PTEN inhibitor experiments are too strongly worded. Although there are increases in phosphorylated AKTSer473, there is no total AKT blot shown in all of Figure 5. Total AKT is the control that needs to be used to ascertain differences in phosphorylation signal and needs to be shown for each of the immunoblots depicted in Figure 5. Furthermore, there is not much difference in phosphorylated AKT signal in the SF1670-treated cells. When the total AKT immunoblot is performed, densitometric analyses should be performed to see how the ratio of phosphorylated AKT to total AKT is changed. Additionally, these results would be greatly improved in conjunction with genetic complementary approaches. These include overexpressing a constitutively active form of AKT or knocking-in phosphatase-deficient PTEN mutations into the RPE-1 hTert, M059K, or U2OS cells. One of these two experiments would need to be performed prior to re-submission.
6) In the Introduction (lines 79-82) and Discussion (lines 390-5), great attention is given to how the results are useful for determining therapeutic options for patients with intact or deficient PTEN. This is overstated given only one human glioblastoma cell line is used in the body of the manuscript. This overstatement needs to be addressed accordingly.
Minor Comments:
7) In the Introduction (lines 55-61), there needs to be some additional citations included for each of the cancers listed or a PTEN review.
8) Please provide similar quantification for Supplementary Figure 2A and 2B as was done for Figure 1B and 1C.
9) The RAD51 immunofluorescence images in Figure 3 are small and hard for the reader to see the foci. Larger and/or brighter images (especially brighter for the channel where RAD51 was imaged) need to be provided for the reviewer/reader.
10) The “suggesting synergistic interactions” claim mentioned at the end of the Results (line 307) needs to be re-worded. Pharmacological synergy can only be measured using isobolograms or using the Combination Index developed by Chou and Talalay. Synergy is a very specific pharmalogical phenomenom. Please augment the words used to describe this potential super-additive effect.
11) Sections 3.3 and 3.4 in the Discussion are titled the exact same. It does not appear that this was intentional. Please rectify.
Author Response
Response to the Review Report 1
“The DNA double-strand break (DSB) repair machinery is critical for maintaining genomic integrity and eliminating genotoxic stresses that could become oncogenic. However, by inducing DNA DSBs via chemotherapy or ionizing radiation in cells that are deficient in their ability to repair these DSBs, tumor cells can be treated and eliminated effectively, albeit for a short time in many cases. Therefore, understanding the molecular mechanisms that underlie how these cells can be made more sensitive to ionizing radiation or therapies targeting these DSB repair pathways will be critical in the development of novel therapeutic regimens. There are four main types of DNA DSB repair pathways – homologous recombination, single-strand annealing (SSA), classical non-homologous end joining, and alternative end-joining. PTEN, one of the most altered tumor suppressors in human cancers, has been shown by other groups to be involved in DNA damage repair, with major attention given to homologous recombination. However, little is known about PTEN’s role in the other major DNA DSB repair pathways. Here, Pei et al show that by losing PTEN expression there is a decrease in homologous recombination repair of DNA DSBs with concomitant increases in the activity of the SSA DNA damage repair pathway. No change in the two end-joining pathways was observed. The SSA DNA damage repair pathway is highly error-prone. The induction of this pathway and the reduction of the homologous recombination pathway is postulated as the mechanism underlying the increased sensitivity of PTEN-depleted cells to DNA damage-inducing ionizing radiation and PARP inhibitors. This work is a step forward in understanding PTEN’s role in regulating DNA DSB repair across all four major DSB repair pathways in retinal epithelial cells, glioblastoma, and osteosarcoma cell lines. However, the impact of the work is dampened because of the use of cell lines whose connection to one another is unclear and deficits in the rigorous use of siRNAs, statistical reporting, and over-interpretation of the results for glioblastomas.”
We thank the reviewer for highlighting the importance of elucidating the mechanisms of DSB repair defects in PTEN-deficient tumor and normal cells. We also greatly appreciate the recognition of our work as “a major step forward” in this field of investigation.
In the revised manuscript we made every effort to address points and concerns raised by the reviewer and explain in detail the rationale of our response. We hope that by doing so we improved the scientific quality of our manuscript.
Major comments:
“1) Pei et al use an immortalized retinal epithelial (RPE-1 hTert), a glioblastoma (M059K), and an osteosarcoma (U2OS) cell line throughout the work. It is not clear how generalizable the results obtained in any one of the three model systems would be for the other cell lines used in this manuscript or for the cancer types that they represent. It is somewhat understandable why U2OS cells were used as they already expressed the DSB repair pathway constructs, it seems a more representative experiment would have been to include these constructs in either the RPE-1 hTert or M059K cell lines as well. Therefore, more impactful results would be obtained if 1) the constructs were added to RPE-1 or M059K and the U2OS results repeated in the newly generated cell lines and if 2) more cancer-specific cell lines – such as a second or third retinal epithelial, glioblastoma, and osteosarcoma cells – were also included. This would increase the generalization of the results, especially with the focus of the Introduction and the Discussion on glioblastoma cells. These experiments would need to be repeated throughout the manuscript with the additional cell lines.”
The suggestion of the reviewer is valid and is required for generating a sound operational background for cancer treatment protocols needed to guide pre-clinical studies based on PTEN status. This ambition is beyond the scope of the present paper, but an important component of our future work. Here, we only define players and concepts that will help to build these further studies. As explicitly indicated in the abstract: “We emphasize consequences of PTEN loss in the engagement of the four DSB repair pathways: classical non-homologous end-joining (c-NHEJ), HR, alternative end-joining (alt-EJ) and single-strand annealing (SSA), and analyze resulting dynamic changes in their utilization.”
We wish to make our findings available to other investigators to guide similar experiments that consider these aspects. The work required to achieve the goals outlined by the Reviewer, which are also our long-term goals, is very extensive and will require input from different laboratories. We also consider the possibility that not all cell lines will respond identically. We have firsthand evidence along these lines from another piece of work we have recently published (Li et al 2020, SCFSKP2 regulates APC/CCDH1-mediated degradation of CTIP to adjust DNA-end resection in G2-phase; Cell Death Dis 2020 Vol. 11 Issue 7 Page 548). In this study, an exhaustive screening identified cell lines with a whole spectrum of responses to the reported effects. Thus, rather than confirming current results with a few more cell lines, we plan to screen a panel of cell lines, establish the spectrum of responses and use the information to define predictors of response. Guided by this long-term strategy, we refrained from extending the experiments in this direction. However, we have revised the paper to alert the readers to these limitations and long-term ambitions.
The U-2 OS cell lines used in our work are well-established and well-characterized DNA repair reporter cell lines, which are routinely and almost universally utilized in the field by several investigators and in diverse experimental contexts to study effects on specific DSB repair pathways. Nearly 90% of papers in the field use this set of cell lines from Dr. Jeremy Stark, suggesting that it is not trivial to generate and characterize such reporter cell lines. We know this actually first-hand, because we extensively tried but only rarely succeeded, and even then, with much lower signal intensities for HR. This is why the generation of reporter cell lines in a different genetic background was not a focus of the present work.
“2) A major drawback of the work is the use of only one siRNA targeting PTEN. This is unacceptable. Multiple siRNAs performed individually or pooled together should be used for these studies. By only using one siRNA in the study, you discount any effects that may be due to off-target effects by the siRNA.”
The reviewer is, of course, absolutely correct for all the studies that have a “discovery” step of siRNA in their evolution. In our case, we took guidance from the literature and selected a siRNA that has been used successfully multiple times (see references provided now). This background and the excellent knockdown achieved at the protein level give us confidence in our results. The representative western blot analysis indeed indicates that the selected PTEN siRNA performs as expected.
“3) The rationale for the use of RPE-1 hTert and M059K having relatively similar levels of PTEN expression is not entirely valid. Based on the information provided in the original immunoblots, these two sets of cell line experiments were not run on the same immunoblot. Even minor alterations in the buffer conditions (from the beginning of the immunoblot with SDS-PAGE running, transfer, blocking, antibody, or wash buffers) or chemiluminescent exposure times can cause inconclusive comparisons between blots. These protein lysates need to be run on the same immunoblot.”
We apologize for the misleading writing. The sentence on similarity was not meant to be taken quantitatively. Indeed, the content of the manuscript is not in any way dependent on the such quantitative comparison. However, as it is indicated in Materials and Methods, the western blot membranes are scanned by a Li-COR infrared scanner, which sensitively detects fluctuations in band intensity. Of course, the reported similarity is supported by several other immunoblots we have. Qualitative intercomparisons are possible in the results presented using the actin control as guidance (Figure 1 and Figure 5). We have revised the passage to avoid this confusion for the readers.
“4) Although most of the Materials and Methods section is written well and in-depth, there are major issues with two aspects, detailed below.
- There is no clearly defined Statistical Analysis section. There needs to be dedicated space to list all the statistical tests used in the paper. The only test noted in the manuscript was Student’s t test. That needs to be rectified. Furthermore, only the quantification of the flow cytometry plots in Figure 2 have p-values. A statistical test needs to be performed to display a p-value for the reader for multiple Figures (1C, 1D, 3B, 3D, 4B, 4D, 4E, 4F, 6-8 (all panels), and S1D).
We agree with the Reviewer and apologize for the serious omission. In the revised manuscript, we include the complete statistical analysis of our Results. The approaches used for this purpose are described in a separate section under Material and Methods
b.The cell lines need to be authenticated, via methods such as STR analysis, to confirm that the cells that are used for the experiments are what they say they are. Furthermore, there is no mention of mycoplasma testing. Mycoplasma-positive cultures can produce erroneous results across a variety of experimental readouts. Testing for mycoplasma and verification that the cell lines are mycoplasma negative will be needed.”
The cell lines utilized in the current study are used in several other projects and have been authenticated in the past.
We also routinely run mycoplasma tests in the laboratory. Moreover, for cell lines we irradiate at outside facilities, which are also included in the present paper, we are required to provide mycoplasma tests run by an independent third-party service. We are therefore confident that our cells are mycoplasma free. We indicate this in the revised manuscript.
“5) The conclusions from the PTEN inhibitor experiments are too strongly worded. Although there are increases in phosphorylated AKTSer473, there is no total AKT blot shown in all of Figure 5. Total AKT is the control that needs to be used to ascertain differences in phosphorylation signal and needs to be shown for each of the immunoblots depicted in Figure 5. Furthermore, there is not much difference in phosphorylated AKT signal in the SF1670-treated cells. When the total AKT immunoblot is performed, densitometric analyses should be performed to see how the ratio of phosphorylated AKT to total AKT is changed. Additionally, these results would be greatly improved in conjunction with genetic complementary approaches. These include overexpressing a constitutively active form of AKT or knocking-in phosphatase-deficient PTEN mutations into the RPE-1 hTert, M059K, or U2OS cells. One of these two experiments would need to be performed prior to re-submission.”
We agree with the reviewer that the total level of the unphosphorylated AKT protein is a better representation for the calculation of the differences in AKT S473 phosphorylation after treatment with selected PTEN inhibitors. In the revised version of the manuscript, we quantified by densitometry the level of the investigated proteins and have included the quantification data in Figure S4. This experiment provides a qualitative assessment that the PTEN inhibitors utilized indeed work - which is sufficient for the conclusions we wish to draw. Further experiments are beyond the scope of the present study.
“6) In the Introduction (lines 79-82) and Discussion (lines 390-5), great attention is given to how the results are useful for determining therapeutic options for patients with intact or deficient PTEN. This is overstated given only one human glioblastoma cell line is used in the body of the manuscript. This overstatement needs to be addressed accordingly.
We have edited the corresponding section to alert the reader that more work will be needed before decisions informing the clinical application of the information can be made.
Minor Comments:
“7) In the Introduction (lines 55-61), there needs to be some additional citations included for each of the cancers listed or a PTEN review.”
We include additional references as suggested.
“8) Please provide similar quantification for Supplementary Figure 2A and 2B as was done for Figure 1B and 1C.”
The requested quantification is added in the corresponding figures.
“9) The RAD51 immunofluorescence images in Figure 3 are small and hard for the reader to see the foci. Larger and/or brighter images (especially brighter for the channel where RAD51 was imaged) need to be provided for the reviewer/reader.”
We thank the Reviewer for pointing out this limitation. We exchanged original images with better ones, which include the EdU channel used for cell cycle-dependent analysis.
“10) The “suggesting synergistic interactions” claim mentioned at the end of the Results (line 307) needs to be re-worded. Pharmacological synergy can only be measured using isobolograms or using the Combination Index developed by Chou and Talalay. Synergy is a very specific pharmalogical phenomenom. Please augment the words used to describe this potential super-additive effect.”
We agree with the Reviewer and re-phrased the passage accordingly.
“11) Sections 3.3 and 3.4 in the Discussion are titled the exact same. It does not appear that this was intentional. Please rectify.”
We apologize for this mistake. We have corrected the titles.

Reviewer 2 Report
The article “PTEN loss enhances error-prone DSB processing and tumour cell radiosensitivity by suppressing RAD51 expression and homologous recombination” by Xile Pei and colleagues investigates the effect of PTEN depletion in cell lines viability. It also tries to elucidate the mechanism underlying the process.
The questions the authors try to answer are of fundamental importance, but this reviewer’s opinion is that the experimental design is flawed and does not provide enough scientific data to support the conclusion.
The main objections are:
1) The three cell lines used in the research are from different “background” with no commonalities.
2) The Results 2 are quite unclear, and possibly there is a typo in line 141-142 when indicating the Figure number relative to the statement.
3) The immunoblotting results in Results 3 are unclear with no distinctive pattern; quantification might be helpful though no clear differences can be detected by eye.
4) Results 5 lacks in additional data and statistical analysis is missing.
5) There is really no clear difference between the siNC and the siPTEN, when comparing the relative untreated and the Olaparib/BMN673 results, with only the higher radiation doses data points partially significant. Moreover, statistical analysis is needed.
Finally, the fact that the combination of radiation plus therapeutics has the same effect on normal and tumour cells indicates the radiosensitization effect is not specific for cancer cells, making the entire conclusion invalid.
Author Response
Response to the Review Report 2
“The article “PTEN loss enhances error-prone DSB processing and tumour cell radiosensitivity by suppressing RAD51 expression and homologous recombination” by Xile Pei and colleagues investigates the effect of PTEN depletion in cell lines viability. It also tries to elucidate the mechanism underlying the process.
The questions the authors try to answer are of fundamental importance, but this reviewer’s opinion is that the experimental design is flawed and does not provide enough scientific data to support the conclusion.”
We thank the reviewer for recognizing the importance of the topic of our paper. We hope however that the Reviewer will find the revised version more convincing than the original one, particularly after the explanations and justifications we provide below.
“1) The three cell lines used in the research are from different “background” with no commonalities.”
As we point out also in our response to point 1 of Reviewer 1, the study aims to identify operational space for repair pathway balance in a PTEN deficient background, in cells genetically “normal” for PTEN that are made deficient by knockdown of the protein. In this regard, the cells on which we base our comparisons are indeed isogenic, with only difference in the levels of PTEN expression (modulated by siRNA knock-down). Our conclusions are NOT based on comparisons between the cell lines used – although the results with all cell lines are similar in their trends. Therefore, their non-isogenic character does not seem as a fatal limitation. In our future work, as we outline above, we plan to cover, using more cell lines, additional aspects of PTEN function, including tumor cell lines with lost PTEN, different levels of expression etc. As we also discuss in the paper, the approach employed eliminates the need to complement PTEN loss by expressing PTEN from extrachromosomal DNA vectors, at levels that can be controlled only with difficulty.
“2) The Results 2 are quite unclear, and possibly there is a typo in line 141-142 when indicating the Figure number relative to the statement.”
We thank the reviewer for pointing out these limitations. We corrected and extended the passage, clarified the description of the results summarized in Figure 2 and under Supplementary information, and cleaned the paper from inconsistencies that came to our attention. We also discuss Figure S1D, which illustrates radiosensitization of the U-2 OS cell line after PTEN knock-down.
“3) The immunoblotting results in Results 3 are unclear with no distinctive pattern; quantification might be helpful though no clear differences can be detected by eye.”
We added selective quantification of the immunoblot images, presented in Figure 5 to support the key points made. The quantitative densitometry data is now included as Figure S4 in the revised version of the manuscript.
“4) Results 5 lacks in additional data and statistical analysis is missing.”
We include now statistical analysis for all generated experiments and describe the results more extensively.
“5) There is really no clear difference between the siNC and the siPTEN, when comparing the relative untreated and the Olaparib/BMN673 results, with only the higher radiation doses data points partially significant. Moreover, statistical analysis is needed.”
We added statistical analysis that allows evaluation of significance as a function of radiation dose.
“Finally, the fact that the combination of radiation plus therapeutics has the same effect on normal and tumour cells indicates the radiosensitization effect is not specific for cancer cells, making the entire conclusion invalid.”
When PARP1 inhibitors are used systemically as monotherapy, it is essential to have minimal effect on normal tissue. However, when irradiation is involved the situation is different, because it is given in a highly localized manner. ONLY normal tissue within that radiation field will be affected to a similar extent as the tumor. The vast majority of remaining normal tissue will not be affected at all. This suggests that the combination may still be useful. Of course, this will need to be investigated in detail. We make of short notice on this at the end of Discussion.

Reviewer 3 Report
In the manuscript "PTEN loss enhances error-prone DSB processing and tumor cell radiosensitivity by suppressing RAD51 expression and homologous recombination" the authors attempt to determine a mechanistic function for PTEN in HR repair, this being expression of RAD51.
The this manuscript is clearly written and easy to understand.
The authors clearly examine why each cell line was used and when possible use multiple cell lines for experiments.
The data is clear and nicely presented in a logical manner.
The proper controls are used to enhance data.
One concern for this reviewer is that the imagining data is not clear. The nuclear staining dominates making it hard to see the foci being measured.
Author Response
Response to the Review Report 3
“In the manuscript "PTEN loss enhances error-prone DSB processing and tumor cell radiosensitivity by suppressing RAD51 expression and homologous recombination" the authors attempt to determine a mechanistic function for PTEN in HR repair, this being expression of RAD51.”
“The manuscript is clearly written and easy to understand.”
We are grateful to the Reviewer for his support and for the recognition of the value of our contribution to the field.
“The authors clearly examine why each cell line was used and when possible use multiple cell lines for experiments.”
We thank the reviewer for the comment. Indeed, the cell lines were selected as explained in the manuscript, and we are confident with our conclusions because cell lines from different genetic backgrounds show similar responses, qualitatively.
“The data is clear and nicely presented in a logical manner.”
We appreciate the comment.
“The proper controls are used to enhance data.”
We thank the reviewer for making this point.
“One concern for this reviewer is that the imagining data is not clear. The nuclear staining dominates making it hard to see the foci being measured.”
We thank the Reviewer for this valid point of critique as well. We have now replaced all flagged images with better images that clearly make the point.

Round 2
Reviewer 1 Report
Thank you for re-submitting this article. The author’s provide a thorough point-by-point rebuttal to the concerns I had following the first submission. I found the rebuttal mostly addressed my previous comments concerns. The rationale provided for the cell lines that were used in Comment #1 was presented, and the rationale makes sense within the larger scope and narrative of the manuscript. I do appreciate the author’s word choices used to address my concerns from major comments 3 and 6. Furthermore, the changes to the Materials and Methods to rectify the comments I had about statistical analyses and cell culture authentication are sufficient (Major comment #4). As far as major comment #5, I do hope the authors try some of those experiments for future manuscripts, and I concede that the experimental work for Major Comment #5 would be beyond the scope of this manuscript. The minor comments were sufficiently addressed.
As for comment #2, I am still concerned that one siRNA is not sufficient for this present work, and the only major point where I disagree with the author’s rebuttals. The conclusions presented in the manuscript would be all the much more rigorously verified if a second siRNA or pooled siRNA targeting PTEN was used. I will concede that an exhaustive re-doing of the manuscript with an entire second siRNA set is beyond the scope of the manuscript as presented, but a targeted follow-up with these additional siRNAs to repeat Figures such as 1D or 2A and 2B still needs to be completed.
Author Response
Response to the Reviewer’s comments
IJMS
“PTEN loss enhances error-prone DSB processing and tumor cell radiosensitivity by suppressing RAD51 expression and homologous recombination”
Pei et al.
ijms-1951203
Response to the Review Report 1
“Thank you for re-submitting this article. The author’s provide a thorough point-by-point rebuttal to the concerns I had following the first submission. I found the rebuttal mostly addressed my previous comments concerns. The rationale provided for the cell lines that were used in Comment #1 was presented, and the rationale makes sense within the larger scope and narrative of the manuscript. I do appreciate the author’s word choices used to address my concerns from major comments 3 and 6. Furthermore, the changes to the Materials and Methods to rectify the comments I had about statistical analyses and cell culture authentication are sufficient (Major comment #4). As far as major comment #5, I do hope the authors try some of those experiments for future manuscripts, and I concede that the experimental work for Major Comment #5 would be beyond the scope of this manuscript. The minor comments were sufficiently addressed.
We appreciate the positive evaluation by the Reviewer of our revision and the understanding shown to our responses. We hope that the revisions made to address points raised improved the manuscript and preempt similar readers’ inquiries.
As for comment #2, I am still concerned that one siRNA is not sufficient for this present work, and the only major point where I disagree with the author’s rebuttals. The conclusions presented in the manuscript would be all the much more rigorously verified if a second siRNA or pooled siRNA targeting PTEN was used. I will concede that an exhaustive re-doing of the manuscript with an entire second siRNA set is beyond the scope of the manuscript as presented, but a targeted follow-up with these additional siRNAs to repeat Figures such as 1D or 2A and 2B still needs to be completed”
We actually were not intending to give the impression that we only tried one siRNA. We just felt that it will add disproportionally to the work load and costs if we carried out all experiments with several siRNAs. Now that the reviewer clarified the request we expanded the re-revised manuscript to include as supplementary information the experiments that helped us to select the siRNA finally used throughout. For this purpose, we incorporated into the manuscript our initial screening utilizing a set of three PTEN specific siRNAs (siRNA-A, siRNA-B and siRNA-C; the RNA sequences of all siRNA are presented in the revised Materials and Methods). Western blot results of the first experiment show a similar degree of protein knockdown after transfection of RPE-1 hTert cells with siRNA-A and siRNA-B and almost no effect when siRNA-C was utilized. The second experiment confirms the equal effectiveness of siRNA-A and siRNA-B and forms the foundation of our selection. The new data is now incorporated in Figure S1.
Unfortunately, we did not carry forward both siRNAs in the experiments shown thereafter. It will take time to do this, as the principal author of the paper returned to his clinical duties in his country a couple of months ago, and running the experiments with repeats with different scientists will take months. We feel the consistency of the data gained with siRNA-B throughout the paper gives assurance to its validity. We also feel that the benefits of a timely publication outweigh the uncertainty generated by the use of a single siRNA, particularly because our work is not the first, or one of the first, to knockdown PTEN. Indeed numerous papers in the literature deal with the topic. We sincerely hope that the Reviewer will give us this leeway.

Reviewer 2 Report
This reviewer thanks the authors for trying and submit an update version of the manuscript. However, it is this review opinion that the manuscript has not been improved with the modifications provided by the authors.
Point 1: "the study aims to identify operational space for repair pathway balance in a PTEN deficient background". If this is the main aim of the paper, three cell lines are not an adequate number to support such research. More cell lines need to be added to draw such important conclusion.
Point 3: the quantification of the WB data needs to be included in the main manuscript, under each blot line, to demonstrate the clear difference in the results, which is still this reviewer opinion is not there.
Point 4: "Statistical analysis was generated by the online version of the MedCalc", the authors need to specify which statistical analysis they have conducted as MedCalc is just a general software.
Point 5: statistical analysis needs to be added in the figure, which once again does not show a clear difference in this reviewer's opinion.
Point 6: "ONLY normal tissue within that radiation field will be affected to a similar extent as the tumor. The vast majority of remaining normal tissue will not be affected at all." It looks like the effect is quite evident, while it should be much less compared to the tumor. Moreover, claiming the surrounding health tissue would be spared without experimental data to support it is a scientific discrepancy.
Author Response
Response to the Reviewer’s comments
IJMS
“PTEN loss enhances error-prone DSB processing and tumor cell radiosensitivity by suppressing RAD51 expression and homologous recombination”
Pei et al.
ijms-1951203
Response to the Review Report 2
“This reviewer thanks the authors for trying and submit an update version of the manuscript. However, it is this review opinion that the manuscript has not been improved with the modifications provided by the authors.”
Respectfully, we think that we made numerous changes improving the manuscript.
“Point 1: "the study aims to identify operational space for repair pathway balance in a PTEN deficient background". If this is the main aim of the paper, three cell lines are not an adequate number to support such research. More cell lines need to be added to draw such important conclusion.”
We kindly refer the Reviewer to our previous arguments.
“Point 3: the quantification of the WB data needs to be included in the main manuscript, under each blot line, to demonstrate the clear difference in the results, which is still this reviewer opinion is not there.”
We have implemented this request in the revised version of the Figure 5. The quantitative data shown by bar plots in Figure S4 is now also included as numbers in the corresponding figure.
“Point 4: "Statistical analysis was generated by the online version of the MedCalc", the authors need to specify which statistical analysis they have conducted as MedCalc is just a general software.”
As we indicate in the “Statistical analysis section” of the revised manuscript we used: MedCalc Software (MedCalc Software Ltd. Comparison of means calculator. https://www.medcalc.org/calc/comparison_of_means.php (Version 20.115; accessed October 5, 2022))
“Point 5: statistical analysis needs to be added in the figure, which once again does not show a clear difference in this reviewer's opinion.”
The included statistical analysis should allow the reader to see which of the differences shown reach statistical significance. Indeed, this is not the case for all doses.
“Point 6: "ONLY normal tissue within that radiation field will be affected to a similar extent as the tumor. The vast majority of remaining normal tissue will not be affected at all." It looks like the effect is quite evident, while it should be much less compared to the tumor. Moreover, claiming the surrounding health tissue would be spared without experimental data to support it is a scientific discrepancy.”
We say that cells outside the radiation field, where the vast majority of the normal tissue is located will not experience adverse effects from radiation despite the fact that all normal tissues in a patient will “see” the inhibitor, if it circulates in the blood. We consider this as a benefit. The aspect of normal tissue protection is the basis of the entire field of treatment planning in Radiation Oncology.

Round 3
Reviewer 2 Report
Although not entirely convinced, this review accept the manuscript in its present form.
Author Response
Response to the Review Report 2
IJMS
“PTEN loss enhances error-prone DSB processing and tumor cell radiosensitivity by suppressing RAD51 expression and homologous recombination”
Pei et al.
ijms-1951203
"Although not entirely convinced, this review accept the manuscript in its present form."
We appreciate the positive decision of the Reviewer.